# Transcriptional and Metabolic Characterization of Feeding Ramie Growth Enhanced by a Combined Application of Gibberellin and Ethrel

**DOI:** 10.3390/ijms231912025

**Published:** 2022-10-10

**Authors:** Hongdong Jie, Yushen Ma, De-Yu Xie, Yucheng Jie

**Affiliations:** 1College of Agronomy, Hunan Agricultural University, Changsha 410128, China; 2Hunan Provincial Engineering Technology Research Center for Grass Crop Germplasm Innovation and Utilization, Changsha 410128, China; 3Department of Plant and Microbial Biology, North Carolina State University, Raleigh, NC 27695, USA

**Keywords:** GA_3_, ETH, transcriptomics, metabolomics, secondary metabolites

## Abstract

Feeding ramie cultivars (*Boehmaria nivea* L.) are an important feedstock for livestock. Increasing their biomass and improving their nutritional values are essential for animal feeding. Gibberellin (GA_3_) and ethylene (ETH) are two plant hormones that regulate the growth, development, and metabolism of plants. Herein, we report effects of the GA_3_ and ETH application on the growth and plant metabolism of feeding ramie in the field. A combination of GA_3_ and ETH was designed to spray new plants. The two hormones enhanced the growth of plants to produce more biomass. Meanwhile, the two hormones reduced the contents of lignin in leaves and stems, while increased the content of flavonoids in leaves. To understand the potential mechanisms behind these results, we used RNA-seq-based transcriptomics and UPLC-MS/MS-based metabolomics to characterize gene expression and metabolite profiles associated with the treatment of GA_3_ and ETH. 1562 and 2364 differentially expressed genes (DEGs) were obtained from leaves and stems (treated versus control), respectively. Meanwhile, 99 and 88 differentially accumulated metabolites (DAMs) were annotated from treated versus control leaves and treated versus control stems, respectively. Data mining revealed that both DEGs and DAMs were associated with multiple plant metabolisms, especially plant secondary metabolism. A specific focus on the plant phenylpropanoid pathway identified candidates of DEGs and DEMs that were associated with lignin and flavonoid biosynthesis. Shikimate hydroxycinnamoyl transferase (HCT) is a key enzyme that is involved in the lignin biosynthesis. The gene encoding *B. nivea* HCT was downregulated in the treated leaves and stems. In addition, genes encoding 4-coumaryl CoA ligase (4CL) and trans-cinnamate 4-monooxygenase (CYP73A), two lignin pathway enzymes, were downregulated in the treated stems. Meanwhile, the reduction in lignin in the treated leaves led to an increase in cinnamic acid and p-coumaryl CoA, two shared substrates of flavonoids that are enhanced in contents. Taken together, these findings indicated that an appropriate combination of GA_3_ and ETH is an effective strategy to enhance plant growth via altering gene expression and plant secondary metabolism for biomass-enhanced and value-improved feeding ramie.

## 1. Introduction

Ramie (*Boehmaria nivea* L.) is a perennial herb. It is not only an important traditional fiber crop but also an appropriate feedstock with a high content of total protein for livestock [1]. The stems and leaves of feeding ramie plants contain 22% crude protein, 1.02% lysine, and 4.07% calcium [2]. It was reported that when alfalfa (*Medicago sativa*) feed (85%) was mixed with ramie (15%) to feed rabbits, the mixed feed was observed to improve the growth performance of rabbits [3]. When a ramie silage was used as a substitute for alfalfa hay to feed cows, the production performance, milk composition, and serum index of cows were improved [4]. These data indicate that protein-rich ramie feed is a type of high-quality forage. Although these advantages show the significance of the nutritional values of ramie feed for animals, this plant has certain negative features that affect the taste of the feed and its digestibility. These disadvantages are caused by lignin, strong fibers, and plant secondary metabolites. Accordingly, the contents of lignin and fibers and the composition of plant secondary metabolites need to be optimized to overcome these problems for high-quality ramie feed.

Lignin is a class of complex water-insoluble polymers of phenylpropanoids. It is an important component of plant cell walls and plays an important role in the mechanical support, water transport, and defense of plants [5]. However, a high content of lignin in forage crops negatively affects the digestibility of feeds for animals and the absorption of feed nutrients by ruminants, thus reduce the nutritional value of forage and feed crops [6]. To date, to solve this problem, multiple studies have been completed to reduce lignin contents in plant tissues without negative impacts on plant growth. One report has indicated that an appropriate experimental design can proportionally reduce in lignin content but does not affect plant growth and biomass [7]. This datum shows that promoting a lignin reduction in forage crops is good for animal husbandry [8].

Plant hormones, such as indole acetic acid (IAA), gibberellin A_3_ (GA_3_), and ethylene (ETH), not only play essential roles in plant growth and development but also largely affect plant metabolism [9]. Given that lignin is a main structure of plants, the effects of plant hormone on its biosynthesis has gained multiple investigations for different purposes. Different plant hormones have been reported to alter the formation and contents of lignin monomers and lignin via impacting gene expressions involved in their biosynthetic pathways [10,11,12,13]. IAA and GA_3_ were reported to alter lignin biosynthesis in mint (*Mentha* spp.) [14]. An exogenous application of IAA or GA alone was observed to alter the composition of lignin monomers in the xylem and phloem. IAA could significantly reduce the ratio of S/G in xylem, while a high concentration of IAA could increase the S/G ratio in the phloem. In addition, a high concentration of GA_3_ decreased the S/G ratio. It was interesting that different combinations of IAA and GA_3_ were observed to affect the degree of polymerization and lignin contents in the phloem but not in the xylem. These data indicate the potential application of IAA or GA_3_ to forage crops for reduction of lignin and increase of digestibility. Plant hormones have been reported to regulate the biosynthesis of flavonoids. GA_3_ was shown to alter the biosynthesis of anthocyanin in plants [15]. An application of exogenous GA_3_ was reported to affect flavonoid accumulation in ginkgo (*Gingko biloba*) leaves [16]. Spraying GA_3_ onto artichokes (*Cynara cardunculus* var. scolymus L.) was found to significantly increase the content of chlorogenic acid [17]. Ethylene was reported to promote the L-phenylalanine ammonia lyase (PAL) activity that led to an enhanced biosynthesis of flavones [18]. During fruit ripening, ETH was found to regulate the gene expression involved in flavonoid metabolism [19,20]. When exogenous ETH alone was applied to grapes (*Vitis vinifera* L.), it was reported to induce the gene expression of anthocyanin biosynthesis [21]. In addition, ETH together with other plant hormones can synergistically alter gene expression. A combination of GA_3_ and ETH was reported to synergistically stimulate the growth of the ectoderm and first leaf of rice (*Oryza sativa* L.) seedlings [22]. Moreover, such synergistic effects of GA_3_ and ETH on the growth of rice seedlings were stable [23]. Taken together, these data indicate that an application of exogenous plant hormones to crops is a useful approach to improving the quality and quantity of products for better values.

Feeding ramie plants are a type of important feedstock for livestock. Reduction in the lignin contents and increase in the biomass have numerous advantages for develop high-quality products with better nutritional values. Herein, we report an application of GA_3_ and ETH to feeding ramie plants grown in the field. A combination of the two hormones affected feeding ramie growth and biomass in the field. To understand potential molecular mechanisms, we completed transcriptomics and metabolomics to characterize both transcriptional and metabolic alterations resulted from the application of the two plant hormones. These data provide useful information to improve the quality and yield of feeding ramie plantation for feedstock.

## 2. Results

### 2.1. Effects of GA_3_ and ETH on the Plant Height and Stem Diameter of Ramie Plants

After the application of GA_3_ and ETH for 25 days, the plant height and stem diameter of feeding ramie were recorded in details. The plant height and stem diameter of the treated groups were significantly higher than those of the control groups. Among them, the average plant height of the treated groups was 79.60 cm, 12.11% higher than that of the control group, 71 cm (Figure 1A). The stem diameter of the treated groups were 10.02 mm, 17.61% bigger than that of the control groups, 8.51 mm (Figure 1B). These data indicated that the application of GA_3_ and ETH significantly increased the vegetative growth.

### 2.2. Influence of GA_3_ and ETH on the Contents of Lignin and Total Flavonoids in the Leaves and Stems of Treated Feeding Ramie Plants

The effects of the GA_3_ and ETH treatment on the contents of lignin and total flavonoids in the leaves and stems of ramie were studied. Compared with the control group plants, the lignin contents were significantly decreased in the treated leaves and stems (Figure 2A,C). The lignin contents in the hormone-treated leaves and stems were about 10.09% and 14.25% (dw), respectively. The lignin contents in the control leaves and stems were 15.6% and 17.51% (dw). These data indicated that the treatment with GA_3_ and ETH decreased the contents of lignin in leaves and stems by 35.3% and 18.6%, respectively. In contrast, the GA_3_ and ETH treatment increased the total flavonoid content in the treated leaves (Figure 2B). The contents of total flavonoids in the treated leaves was 5.22 mg/g (dw). The content of total flavonoids in the control leaves was 3.64 mg/g (dw). These data indicated that the treatment of GA_3_ and ETH significantly increased the total flavonoid content by 43.4% in leaves (Figure 2B). In addition, these data showed that the contents of total flavonoids in the stems slightly responded to the GA_3_ and ETH treatments.

### 2.3. Transcriptome Analysis

#### 2.3.1. Transcriptome Sequencing

Samples from GA_3_ and ETH treated plants (labeled as C) and control group plants (labeled as CK) were sequenced using an Illumina NovaSeq 6000 platform (Illumina, San Diego, CA, USA). A total of 7.51–8.55 billion bases (unit: bp) were obtained from three biological replicates of treated leaves (labeled as C-Y). A total of 6.81–7.79 billion bases (unit: bp) were obtained from three biological replicates of treated stems (labeled as C-J). A total of 5.67–7.05 billion bases was obtained from three biological replicates of control leaves (labeled as CK-Y) and a total number of 5.78–7.49 billion bases were obtained from three biological replicates of control stems (labeled as CK-J). After filtration and trimming, 5.63–7 billion, 6.81–7.79 billion, 7.45–8.49 billion, and 5.74–7.444 billion bases were obtained from C-Y, C-J, CK-Y, and CK-J, respectively. In all the groups, Q20 and Q30 values were 97% and 93%, respectively. The average GC contents were about 48% (Table 1).

#### 2.3.2. Differential Expression Gene (DEG) Analysis

The Spearman correlation coefficient method was used to analyze the degree of biological repeatability between the samples. Based on the reads per kilobase of transcripts per million mapped reads (RPKM) in different samples, the resulting data could reveal the correlation of levels of gene expression between the samples and show a perfect Spearman correlation. The expression levels of genes had twofold changes with a |log2 (fold change)| > 1 value and a *p* < 0.05 value indicated a differential expression between the control and treated samples. As a result, 1562 DEGs were identified between treated and control leaves. Of them, 878 were upregulated in treated leaves, while 684 were downregulated. Meanwhile, 2364 DEGs were identified between treated and control stems. Of them, 792 were upregulated in treated stems, while 1572 were downregulated (Appendix A). These DEGs indicated that the GA_3_ and ETH treatment affected the gene expression in both leaves and stems.

#### 2.3.3. GO Function and Enrichment Analysis of DEGs via the KEGG Pathway Database

The Gene Ontology (GO) database provides an overview of gene functions in molecular function, cellular component, and biological process information. The results of a GO annotation analysis revealed the number of DEGs associated with different functions. Based on the gene number, the treatment of GA_3_ and ETH resulted inDEGs involved in cell processes, metabolic processes, single organism processes, cells, cell sites, organelles, binding sites, and catalytic activities in leaves (Figure 3A, Appendix A). The resulting data showed that the treatment of GA_3_ and ETH resulted in DEGs involved in cellular processes, metabolic processes, single organism processes, cells, cell sites, organelles, binding sites, and catalytic activities in stems (Figure 3B, Appendix A). These data were informative for focusing on gene of interests. 

GO function enrichment analysis was performed to further understand the profiles of the DEGs caused by the GA_3_ and ETH treatment. The result showed that the first five GO terms with significant enrichment of DEGs between control (CK-Y) and treated (C-Y) leaves were redox activity (GO:0016491), redox process (GO: 0055114), ADP binding (GO: 0043531), heme binding (GO: 0020037), and tetrapyrrole binding (GO: 0046906) (Figure 4A). The top five GO terms between the control (CK-J) and treated (C-J) stems heme binding (GO: 0020037), tetrapyrrole binding (GO: 0046906), oxidoreductase activity, acted on paired donors, and bound or reduced molecular oxygen (GO: 0016705), ADP binding (GO: 0043531), and kinesin complex (GO: 0005871). These data indicated the different responses of leaves and stems to GA_3_ and ETH.

DEGs were further metabolically categorized via The Kyoto Encyclopedia of Genes and Genomes (KEGG), a primary public database for pathway annotation. One advantage is that the pathway significance enrichment analysis uses the KEGG pathway as unit to characterize enriched metabolisms. In addition, the hypergeometric test allows determining pathways that are the most significantly enriched in differential genes compared with the whole background. These analyses obtained genes involved in the top 20 enriched metabolisms differentially expressed in treated leaves and stems compared with control ones (Figure 5). The enriched 20 top categories altered in the treated leaves were composed of 120 genes involved in general metabolic pathways, of which 80 were associated plant secondary metabolism. Mainly altered pathways of plant secondary metabolism included phenylpropanoid biosynthesis, sesquiterpene and triterpene biosynthesis, tyrosine metabolism, plant-pathogen interaction, 2-oxocarboxylic acid metabolism, flavonoid biosynthesis, and other pathways (Figure 5A and Appendix A). The enriched 20 top categories altered in the treated stems were composed of 200 genes, of which 120 were featured to associate with plant secondary metabolism, including sesquiterpene and triterpene biosynthesis, brassinosteroid biosynthesis, stilbene, diarylheptane and gingerol biosynthesis, tyrosine metabolism, isoquinoline alkaloid biosynthesis, phenylpropane biosynthesis, flavonoid biosynthesis, and other pathways (Figure 5B and Appendix A). These enriched pathways indicated that on the one hand, leaves and stems shared similar responses to the GA_3_ and ETH treatment; on the other hand, two tissues showed different responses to the two hormones. Of all metabolic alterations, the phenylpropanoid biosynthesis and flavonoid biosynthesis in both the leaves and stems were altered by the GA_3_ and ETH treatments. Genes involved in these two pathways (Appendix A) were selected for expression verification.

#### 2.3.4. Validation of DEGS by Real-Time Quantitative Reverse Transcription PCR (qRT-PCR)

To verify the accuracy of DEG data obtained by RNA-Seq, six DEGs from leaves and the other six from stems (Figure 6) were selected for qRT-PCR analysis. Gene specific primers for fluorescent quantification were designed for amplification of these genes and *B. nivea* Actin (internal control) (Appendix A). The relative level of expression was calculated as described by Livak et al. [24]. The resulting data showed five genes with a higher expression and one with lower expression in treated leaves (C-Y) than control leaves (CK-Y) (Figure 6). These data supported their differential expression levels shown by RNA-seq. In the six DEGs from stems, the expression levels of five and one genes were higher and lower in treated stems (C-J) than in control ones (CK-J). These data supported the results from RNA-seq.

### 2.4. Alteration of Metabolite Profiles by GA_3_ and ETH

#### 2.4.1. Effects of GA_3_ and ETH on Metabolite Accumulations in Leaves and Stems

UTLC-MS/MS was completed to understand the effects of the GA_3_ and ETH treatment on metabolites in leaves and stems. Based on our in-house and publically available metabolite libraries, we could annotate 99 metabolites from the treated and control leaves and 88 metabolites from the treated and control stems. Of leaf metabolites, 44 are flavonoids, and of stem metabolites, 31 are flavonoids. One main group of flavonoids is quercetin derivatives. The levels of these compounds were estimated with UTLC-MS/MS and then were used for principal component analysis (PCA) via publically available R package gmodels. Two ordinate plots for comparisons of both treated vs. control leaves and treated vs. control stems were created to visualize the differentiation of metabolites (Figure 7). Based on the two plots, the accumulation profiles of the 99 compounds were separated between treated (C-Y) and control (CK-J) leaves. Meanwhile, the profiles of the 88 metabolites were apparently separated between treated (C-J) and control (CK-J) stems. These data indicate that the GA_3_ and ETH treatment alters plant metabolisms in both leaves and stems of ramie.

To further understand metabolic differentiations resulted from the GA_3_ and ETH treatment, orthologous projections to latent structural discriminant analysis (OPLS-DA) was completed to evaluate the variable projection importance (VIP) value and the *t*-test *p* value of univariate statistical analysis was calculated to understand statistical significance. The values of VIP ≥ 1 and *t*-test *p* < 0.05 in the OPLS-DA model were used to evaluate statistical significance of metabolite differentiation between treated and control samples. The metabolites in all the leaf samples or stems were normalized by a z-score in rows and then used to develop a heatmap to visualize metabolite differentiation (Figure 8). In the 99 leaf metabolites, the levels of 58 were enhanced while those of 41 were decreased in treated leaves (C-Y) compared with control leaves (CK-Y) (Figure 8A, Appendix A). In the 88 stem metabolites, the levels of 49 were increased, while those of 39 were reduced in the treated stems (C-J) compared with control stems (CK-J) (Figure 8B and Appendix A).

#### 2.4.2. Enrichment Analysis of Differential Metabolites

Differentially accumulated metabolites (DAMs) were submitted to KEGG to map and enrich the metabolic pathways. Two plots were generated with the top 20 enrichments to show metabolites and their pathways enriched by KEGG (Figure 9). In the two plots, the color of the dots represents the Q value of the hypergeometric test. A smaller value indicates a more reliable and statistically significant result. The size of the dot represents the number of differential metabolites in the corresponding pathway. Based on the plots, the DAMs in treated leaves (C-Y) were mapped and enriched in the following six pathways: “flavone and flavonol biosynthesis”, “arginine and proline metabolism”, “phenylalanine metabolism”, “oxidative phosphorylation”, “indole alkaloid biosynthesis”, and “purine metabolism” (Figure 9A, Appendix A). The DAMs in the treated stems (C-J) were primarily mapped to and enriched in the following six pathways: arginine and proline metabolism; glucosinolate biosynthesis; valine, leucine, and isoleucine; flavonoid and flavonol biosynthesis; tryptophan metabolism; and indole alkaloid biosynthesis (Figure 9B, Appendix A). These results indicated that although the treatment of GA_3_ and ETH altered different metabolisms in leaves and stems, the two hormones affected the biosynthesis of flavonoids in the two tissues.

### 2.5. Integration of Differentially Expressed Genes and Accumulated Metabolites Involved in the Phenylpropanoid Pathway

To understand the effects of GA_3_ and ETH treatment on plant metabolisms, we used DEGs, DAMs, and enriched pathways for an integrative analysis. Especially, this analysis focused on the biosynthesis of phenylpropanoids (including flavonoids) that was altered in both leaves and stems treated by the two hormones. This integration created two diagrams to characterize the synergistic alterations of both pathway gene expressions and metabolite levels (Figure 10 and Figure 11). The results showed that the expression of *PAL*, *4CL*, *CCR*, *CAD, C3H,* and *CYP98A* was upregulated in the treated leaves (C-Y) (Figure 10). Directly or indirectly corresponding to these genes’ upregulation, the levels of six intermediates in the phenylpropanoid pathway were increased (Figure 10). For example, the increase of cinnamic acid was associated with the upregulation of *PAL*. These data indicated that the treatment of GA_3_ and ETH led to the upregulation of the phenylpropanoid pathway in leaves. In contrast with the results from the treated leaves, *4CL*, *CYP73A,* and *HCT* were downregulated in their expression levels in the treated stems (Figure 11). The abundance of three intermediates, 3-O-*p*-coumaroylquinic acid, *p*-coumaraldehyde, and cinnamic acid was reduced, while that of neochromogenic acid (5-O-caffeoylquinic acid) was increased (Figure 11). These results indicate that the treatment of GA_3_ and ETH differentially altered the phenylpropanoid pathway in leaves and stems. 

## 3. Discussion

The development of technology is important to improve the biomass of ramie to meet the high demand of feedstock. Unfortunately, engineering of ramie is still in the infant stage because of the difficulty of genetic transformation. In addition, the use of transgenic plants for livestock has strict regulation. Therefore, the goal of our study was to develop alternative technologies to increase the biomass of feeding ramie and to understand plant secondary metabolisms in leaves and stems for valued-improved products. A combined treatment of GA_3_ and ETH was reported to improve the growth of rice seedlings [23]. Based on this and another previous report that exogenously spraying GA_3_ and ETH can promote plant growth and development and alter plant metabolism [25], we hypothesized that these two plant hormones might improve ramie growth, and then selected them to test their effects on the growth of feeding ramie and on plant metabolism in the field of our research station. The experimental data showed that the spraying of two times of GA_3_ and one time of ETH increased plant growth and biomass (Figure 1). This datum indicated the agricultural significance of this treatment for the improvement of ramie biomass. To understand potential molecular and metabolic mechanisms that were likely associated with the enhanced growth, we estimated lignin and total flavonoids in leaves and then employed transcriptomics and metabolomics to characterize transcriptomes and metabolomes. The resulting data revealed the trade-off of enhanced plant growth with the reduction of lignin in both the leaves and stems (Figure 2A,C). This datum supports the fact that the reduction of lignin can increase the plant growth. In contrast with the lignin reduction, the measurement of total flavonoids disclosed different responses of stems and leaves to the two-hormone treatment. The content of total flavonoids was increased in the treated leaves, but not apparently altered in the treated stems (Figure 2 B,D). To understand these observations, we completed RNA-seq and LC-MS/MS analysis (Table 1, Figure 3, Figure 4, Figure 5, Figure 6, Figure 7, Figure 8 and Figure 9). We then mined DEGs and DAMs associated with plant phenylpropanoid biosynthesis. As we hypothesized, the expression of genes and abundance of intermediates involved in the phenylpropanoid pathway were altered by the treatment of GA_3_ and ETH. In particular, the expression of key genes involved in the biosynthesis of lignin was decreased in both treated leaves and stems. Shikimate O-hydroxycinnamoyltransferase (HCT) uses p-coumaroyl-CoA, caffeoyl-CoA, and caffeoyl shikimic acid as substrates in the pathway toward the formation of lignin. The gene encoded HCT was transcriptionally downregulated in treated leaves by GA_3_ and ETH (Figure 10). Although the entry genes such as *PAL* and *4CL* were upregulated to likely increase intermediates such as cinnamic acid, p-coumaric acid and caffeic acid, the downregulation of *HCT* limited the metabolic flux from these substrates to lignin. In the treated stems, *4CL*, *HCT* and *CYP73A* were significantly downregulated by GA_3_ and ETH and three key intermediates were reduced in their abundance. These reductions were directly associated with the reduction of lignin in the treated stems. The reduction of lignin in the treated leaves and stems were supported by multiple previous studies. A study of *HCT* silence reported the reduction of lignin and increase of various flavonoids [26]. The inhibition of HCT in tobacco (*Nicotiana benthamiana*) resulted in significant changes in the amount and composition of lignin [27]. COMT encodes 5-hydroxyconiferol and 5-hydroxyconiferol to produce mustard aldehyde and mustard alcohol, respectively, and is an important enzyme for the synthesis of lignin monomers [28]. Studies reported that downregulating COMT in rapeseed (*Brassica rapa*) reduced the content of lignin [29]. These data show that the integration of transcriptomics and metabolomics can provide molecular evidence to understand the potential mechanisms associated with the enhancement of ramie growth treated by GA_3_ and ETH. More importantly, given the decrease of the lignin content in ramie improve palatability of the leaves [30], the treatment of GA_3_ and ETH shows an importance of application for an improved digestibility [30].

In addition, the development of technologies is important to improve nutritional values of feeding ramie for livestock. Flavonoids are metabolites of interests, because they are phytobiotics benefiting animal health. Our measurement showed the increase of total flavonoids in the treated leaves (Figure 2), which are the main biomass of feedstock. This result was anticipated because GA had been reported to regulate of the flavonoid biosynthesis [31] and ethylene had been shown to promote the formation of flavonoids, such as anthocyanins. Another study reported that a low content of GA_3_ enhanced the biosynthesis of leaf flavonoids and an application of exogenous ETH promoted the accumulation of leaf flavonoids [18]. Our metabolic profiling and transcriptome data provided molecular evidence to support the increase of flavonoids. As described above and shown in Figure 10, the expression of *PAL* and *4CL* was increased and the abundance of cinnamic acid and p-coumaric acid was enhanced in the treated leaves. These two compounds and p-coumaryl CoA are precursors of flavonoids. Due to the reduction of the metabolic flux from these substrates to lignin, one hypothesis was that these two compounds were used to produce flavonoids. p-Coumaryl CoA is the substrate of chalcone synthase, the first committed step to flavonoids. No changes of the p-coumaryl CoA content indicated that this direct substrate of flavonoids was used for the biosynthesis of downstream metabolites. Accordingly, the content of total flavonoids was increased in the treated leaves. Our data were also supported by other studies. A deficiency in *HCT* led to the reduction of lignin and the high production of flavonoids in *A. thaliana* [27]. A *HCT* silence was reported to lead to the reduction of lignin biosynthesis and the enhancement of to the flavonoid biosynthesis [32]. However, contrary to the increase in the treated leaves, the abundance of flavonoids was not altered in the treated stems although the biosynthesis of lignin was downregulated. This likely resulted from the downregulation of *PAL* and *4CL* and the decrease of cinnamic acid in the treated stems. 

## 4. Materials and Methods

### 4.1. Feeding Remie, Field Growth, Treatment with GA_3_ and ETH, Sampling, and Measurement of Plant Height and Stem Diameter

A main feeding ramie variety, namely Zhongzhu No. 1, was used in the experiments. Given that this plant is perennial, they were grown at the Yunyuan experimental station of Hunan Agricultural University (Changsha, China). Three plots were randomly selected from the field and each was used as a plot replicate. The distance of plots were about 10 meters to prevent a potential cross impact in the following plant hormone application experiments. Each plot had more than 100 plants. Each plant was maintained with five vegetative stalks (grown from the rhizomes). For this study, we randomly selected 30 plants from each plot, all of which grew to a similar height and leaf numbers and were free of pests and pathogens. In each plot, the 30 plants were divided in six cells, each of which had 5 plants (about 25 stalks) as one biological sample. To obtain uniform plants for the following experiments, we cut all 90 plants from the basis of stems on 27 July 2021. Then, we fertilized all cut plants with urea and potassium chloride (KCl) on the same day. The amounts of urea and KCl applied for plants are 3.5 kg/100 m^2^ and 2.25 kg/100 m^2^ on the same day after cutting. After this, the plants were not provided additional fertilizer. During the following growth of plants, we observed and recorded regenerated new stalks. On the 10th day after cutting, all plants grew stalks that were approximately 10 cm tall and each plant maintained 5 stalks. We used a ruler to measure all stalks from the base (the ground) to the top of the stem and used a Vernier caliper to measure the stem diameter at the one-third of the plant height from the base to the top. All plants were used for treatment of GA_3_ and ETH described below.

Gibberellin (GA_3_) and ethephon (a product of ETH) were used to treat newly regenerated 10-day old plants. Two products were purchased from Beijing Solepak Technology Co., Ltd. (Beijing, China). We used deionized water (dH_2_O) to prepare 10 mg/L GA_3_ and 5 mg/L ETH to spray plants. Before spraying, six cells (each with 5 plants) in each plot were divided into two groups. One treatment group included three cells, in each of which 15 plants were sprayed with GA_3_ and ETH. The other control group included another three cells, in each of which 15 plants were sprayed with dH_2_O. Three spaying steps were completed to apply GA_3_ and ETH to the treatment group plants. First, new plants were fully sprayed with GA_3_ (fully covered with liquid mist) and then allowed to grow 7 days. Second, all treated plants were given a second spray of GA_3_ as completed for the first one and then allowed to grow another 7 days. Third, treated plants were sprayed with ETH and then allowed to continuously grow 5 days. Meanwhile, all control plants in the control cells were sprayed dH_2_O at same time and allowed to grow the same days as the treated plants. After the treatment, we cut all treated and control plants to collect stems and leaves. Stems and leaves from each cell were pooled together as one biological sample, respectively. Accordingly, each plot had three biological leaf and stem replicates treated by GA_3_ and ETH and three biological control leaf and stem replicates spayed with dH_2_O. In total, we collected 36-pooled biological samples, nine treated stem replicates, nine treated leaf replicates, nine control stem samples, and nine control leaf samples, which were used for different experiments.

After sample collection, each biological sample was divided into two groups. One group was immediately placed in an oven set up at 105 °C for 30 min, and then the temperature was reduced to 65 °C until the weight of sampes was constant. The resulting dry samples were stored in a desiccate chamber under darkness for analysis of lignin and total flavonoids described below. The other group was immediately frozen in liquid nitrogen and then stored in a −80 °C freezer for late transcriptomics and metabolomics experiments described below.

### 4.2. Analysis of Lignin and Total Flavonoids

The dried samples were ground into fine powder in a pulverizer with a screen aperture of 0.2 mm. The powdered samples were sealed in bags, which were stored in a desiccator chamber. The lignin contents in samples were measured with an assay kit (BC4200, Solarbio, Beijing, China) according to the manufacturer’s instructions. The total flavonoids contents in these samples were measured with an assay kit (BC1355, Solarbio) according to the manufacturer’s instructions. The resulting data were expressed as the mean ± standard error from at least three independent experiments. Statistical analysis was performed using a one-way analysis of variance (ANOVA) with SPSS 19.0 (IBM, Inc., Armonk, NY, USA).

### 4.3. RNA-seq and Gene Expression Analysis

RNA extraction, library preparation, high-throughput sequencing, and annotation were completed in Gene Denovo Technology Co., Ltd. (Guangzhou, China). In brief, total RNA was extracted from frozen samples using a TRIzol kit (Invitrogen, Carlsbad, CA, USA) and its quality was assessed using an Agilent 2100 bioanalyzer (Agilent Technologies, Santa Clara, CA, USA). The quality of RNA samples were also examined with RNAase-free agarose gel electrophoresis. To enrich mRNA, rRNA was removed from the total RNA samples using a Ribo Zero^TM^ magnetic Kit (Epicentre, Madison, WI, USA). The resulting mRNA was fragmented into short fragments using fragment buffer and reversely transcribed into cDNA using an NEB Next Ultra RNA Library Preparation Kit for Illumina (NEB#7530; New England Biolabs, Ipswich, MA, USA). The purified double stranded cDNA fragments were end-repaired, base added, and connected to the Illumina sequencing adapters. Ligation reactions were purified with AMPure XP Beads (1.0 X). The linked fragments were sized using agarose gel electrophoresis and amplified using PCR. The resulting cDNA library was sequenced using Illumina NovaSeq 6000. The original sequences were evaluated with FastQC in Trimmomatic version 0.36, by which numerical readings with low-quality were filtered and discarded. Reads contaminated by adapter sequences were cut and removed. The resulting clean reads were assembled with a STATR version 2.5.3a software and mapped to the reference genome of ramie (*B. nivea* L.). All the assembled monogenes were annotated via GO and KEGG database. The values of FPKM were used to estimate the level of gene expression. The differentially expressed genes (DEGs) between treated and control samples were identified using Edger software version 3.12.1. FDR correction *p* < 0.05 and fold change ≥ 2 were used to determine DEGs. The resulting GO and KEGG data were further analyzed using KOBAS software version 2.1.1 with a corrected *p* < 0.05 to determine whether the enrichment was statistically significant. The resulting final DEGs were analyzed with topGO and KEGG paths included in the enrichment analysis version R 3.6.0 (October 2019). 

### 4.4. UPLC-MS/MS Analysis and Metabolite Annotation

#### 4.4.1. Extraction and UPLC-MS/MS Analysis

The leaf and stem samples stored in a freezer were freeze-dried in vacuum and then ground into powder in 1.5 mL tubes placed on Laboratory Mixer Mill MM 400 (Retsch, Haan, Germany) with a zirconia bead at 30 Hz for 1.5 min. The leaves and stems of the treated group plants were labelled as C-Y and C-J. The leaves and stems of the control group plants were labelled as CK-Y and CK-J. For each sample, 100 mg powder was suspended in in 1.0 mL 70% methanol in water contained in 1.5 mL tubes. The tubes were votexed and then were placed at 4 ℃ for overnight. Tubes were centrifuged for 10 min at 10,000× *g*. The supernatant from each tube was transferred to a new one and filtered through a microporous membrane (0.22 μm pore size). The filtered extracts were used for the UPLC-MS/MS analysis.

The UPLC-MS/MS analysis was performed using ultra-performance liquid chromatography (UPLC) (Shimpack UFLC SHIMADZU CBM30A, http://www.shimadzu.com.cn; accessed on/ 7 August 2022) coupled with an Applied Biosystems 6500 QTRAP (a tandem mass spectrometry) (http://www.appliedbiosystems.com.cn/; accessed on 7 August 2022). Metabolites were separated on a Waters ACQUITY UPLC HSS T3 C18 column (1.8 µm, 2.1 mm × 100 mm). The elution solvents included solvent A (0.4% acetic acid in HPLC-MS grade water) and solvent B (99.6% HPLC-MS grade acetonitrile: 0.4% acetic acid). A gradient elution program that was developed to elute metabolites was composed of different ratios of A and B volumes, 95% A: 5% B to 5% A: 95% B (0–11 min), 5% A: 95% B to 95% A: 5% B (11–12 min), 95% A: 5% B (12–12.1 min), and 95% A: 5% B (12.1–15 min). The flow rate was set up at 0.4 mL/min. The injection volume was 2 µL. 

MS analysis was completed with liner ion trap (LIT) and triple quadrupole (QQQ) scans that were acquired on a triple quadrupole-linear ion trap mass spectrometer (Q TRAP), API 6500Q TRAP LC/MS/MS System, equipped with an ESI Turbo Ion-Spray interface operated with the positive ion mode and controlled by Analyst 1.6.3software (AB Sciex). Electrospray ionization (ESI) was completed to generate ions. The ESI source operation parameters were set up for ion source and turbo spray. The source temperature were 500 °C and the ion spray voltage (IS) was 5500V. Ion source gas I (GSI), gas II(GSII) and curtain gas (CUR) were set at 55, 60, and 25.0psi, respectively. The collision gas (CAD) was set at the high mode. Instrument tuning and mass calibration were performed with 10 and 100 μmol/L polypropylene glycol solutions in QQQ and LIT modes, respectively. QQQ scans were acquired as multiple reaction monitoring (MRM) experiments with collision gas (nitrogen) set at 5psi. Declustering potential (DP) and collision energy (CE) for individual MRM transitions was carried out for further DP and CE optimization. A specific set of MRM transitions were monitored for each period according to the metabolites eluted within this period.

#### 4.4.2. Data Pretreatment and Metabolite Identification

Data filtering, peak detection, alignment, and calculation were performed using an Analyst 1.6.1 software. To generate a matrix that excluded potentially biased and redundant data, the peaks of signal/noise (s/n) > 10 were checked manually. A software written in Perl was used to remove redundant signals caused by different isotopes, intra source fragments, K^+^, Na^+^, and NH_4_^+^ adducts and dimers. To facilitate the annotation of metabolites, an accurate m/z value was obtained for each Q1. The total ion chromatograms (TICs) and extracted ion chromatograms (EICS or XICs) of the QC samples were generated, and the metabolite profiles of all the samples were summarized. The area value of each color spectrum peak was calculated. The peaks were arranged in different samples according to their spectral mode and retention time. The metabolites were annotated by searching internal and public databases (Massbank, KNApSAcK, HMDB [28], MoTo DB, and METLIN [33]) and comparing m/z values, retention time, and fragmentation pattern with the standards. The metabolites with the largest difference between the two groups were ranked using the Variable Importance in Projection (VIP) score of the (O) PLS model. The log mean VIP threshold was set as 1. In addition, we used a student’s *t*-test as a single factor analysis to screen for differential metabolites. The *p* < 0.05 and VIP ≥ 1 values in the *t*-test indicated that the metabolites differed between the two groups. KEGG was one of the main public pathway database including not only genes but also metabolites [34]. Those metabolites with a differential accumulation were mapped to the KEGG metabolic pathway for pathway and enrichment analyses. Pathway enrichment analysis could allow understanding differentially expressed pathways that were enriched by the GA_3_ and ETH treatment.

### 4.5. Validation with qRT-PCR

To verify the reliability of transcriptome sequencing results, we randomly selected 12 DEGs for qRT-PCR analysis. Gene specific primers were designed (Appendix A). In addition, actin gene was used as the internal reference. All qRT-PCRs were performed on a Light Cycler 96 system (Roche Diagnostics, GmbH, Mannheim, Germany). Reactions were completed in 20 μL with SYBR Green qPCR Master Mix by following the manufacturer’s manual. The thermal circle was composed of 95 °C for 15 s, 60 °C for 30 s, 72 °C for 30 s, and 40 cycles. The relative expression levels of genes were calculated using the 2^−∆∆CT^ method [24] based on the expression of the actin reference.

### 4.6. Statistical Analysis

All calculated data were expressed as mean ± standard error. All compared data were evaluated with one-way analysis of variance (ANOVA) and then tested for the least significant difference (LSD) using SPSS 26.0 (IBM, Inc., Armonk, NY, USA). Significant difference was accepted at *p*-values < 0.05.

## 5. Conclusions

In conclusion, the use of a combined GA_3_ and ETH is effective to enhance the growth of feeding ramie. GA_3_ and ETH reduce the content of lignin in leaves and stems and increase the contents of total flavonoids in leaves via altering the gene expression of the phenylpropanoid pathway in the two tissues of feeding ramie.

## Figures and Tables

**Figure 1 ijms-23-12025-f001:**
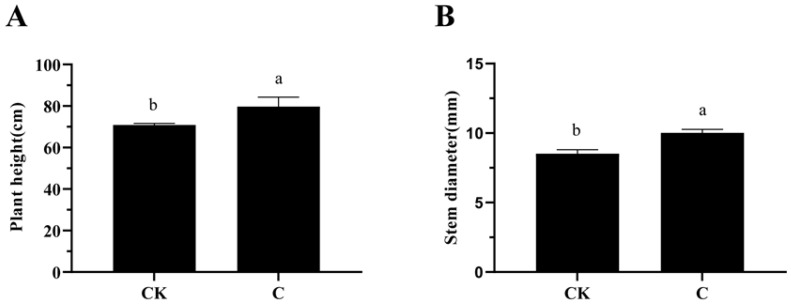
Effect of the GA_3_ and ETH Treatment on Plant Growth. The application of GA_3_ and ETH increased the plant height (**A**) and stem diameter of ramie (**B**). The values on the vertical axis represent the mean ± SE of three biological replicates in three plots. Different letters indicate significant differences at *p* < 0.05. SE, standard error.

**Figure 2 ijms-23-12025-f002:**
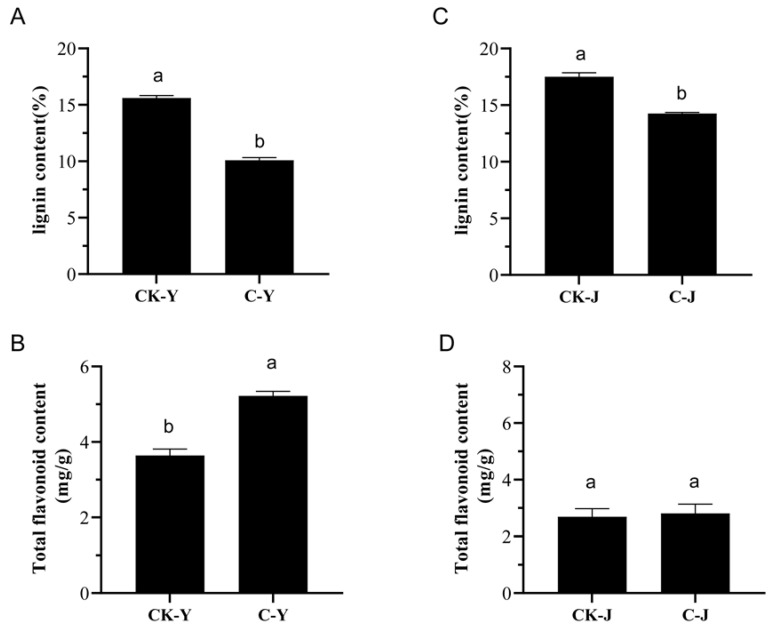
Effects of the GA_3_ and ETH Treatment on the Contents of Lignin and Total Flavonoids in Ramie Leaves (C-y) and Stems (C-J). (**A**,**B**), the contents of lignin (**A**) and total flavonoids (**B**) in treated (C-Y) and control (CK-Y) leaves. (**C**,**D**), the contents of lignin and total flavonoids in treated (C-J) (**C**) and control stems (CK-J) (**D**). The values on the vertical axis represent the mean ± SE of three biological replicates. Different letters indicate significant differences at *p* < 0.05. SE, standard error.

**Figure 3 ijms-23-12025-f003:**
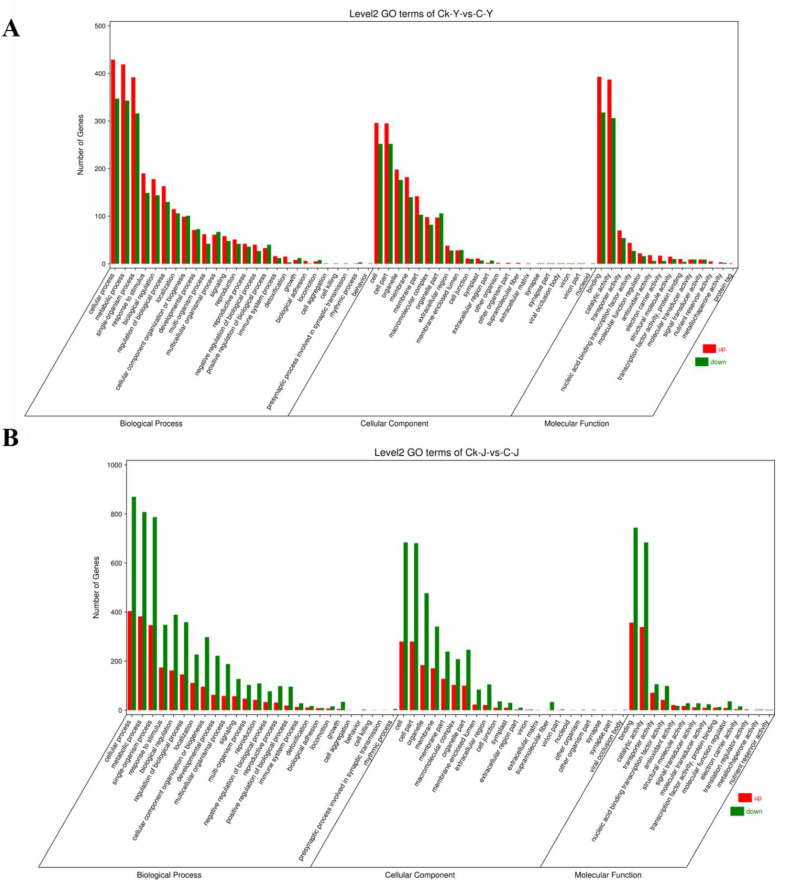
GO Annotation Analysis of the Differentially Expressed Genes (DEGs) between Treated and Control Samples. Two plots show distribution profiles of DEGs between control (CK-Y) and treated leaves (C-Y) (**A**) and between control (CK-J) and treated (C-Y) stems (**B**) in three functional categories. GO enrichment classification histogram: the abscissa is the secondary GO term, and the ordinate is the number of differential genes. Red color indicates upregulation and green color indicates downregulation. GO, Gene Ontology.

**Figure 4 ijms-23-12025-f004:**
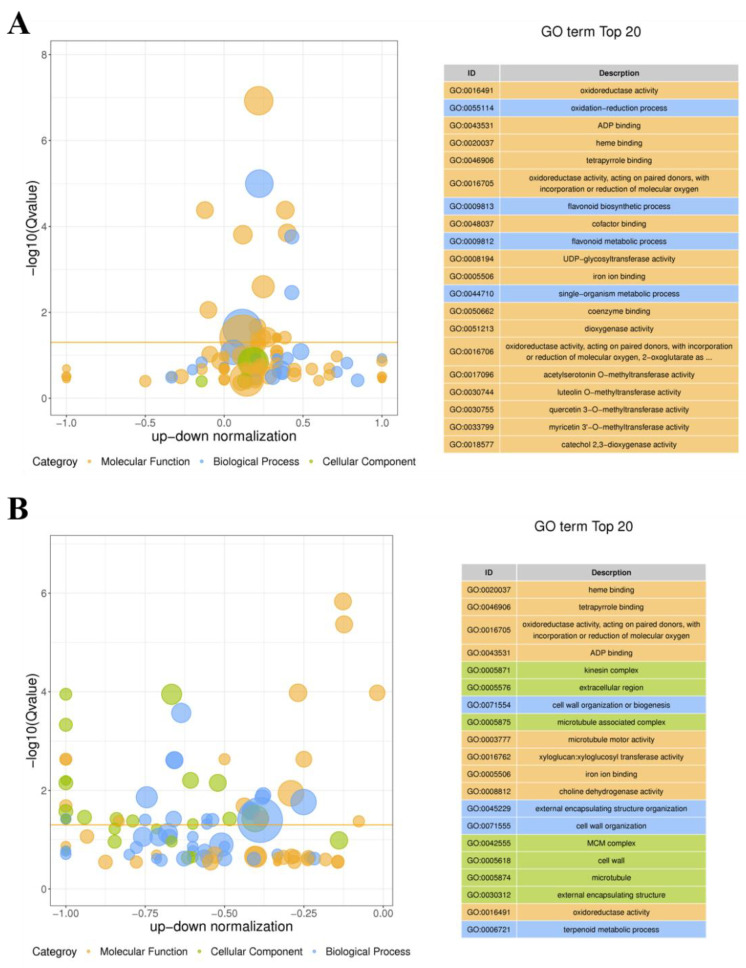
GO Function Enrichment Analysis of Differentially Expressed Genes (DEGs). (**A**,**B**), two charts show the top 20 GO terms of DEGs between control (CK-Y) and treated leaves (C-Y) (**A**) and between control (CK-J) and treated (C-J) stems (**B**). In the GO enrichment difference bubble charts, the ordinate is -log10 (Qvalue) and the abscissa is z-score value. The two charts show the proportion of the difference between the number of differentially upregulated and downregulated genes in the total DEGs. The yellow line represents the threshold of Q value = 0.05. The right tables include the GO term list of the top 20 Q values. Different colors represent different Ontologies, which are separated by their class. GO, Gene Ontology.

**Figure 5 ijms-23-12025-f005:**
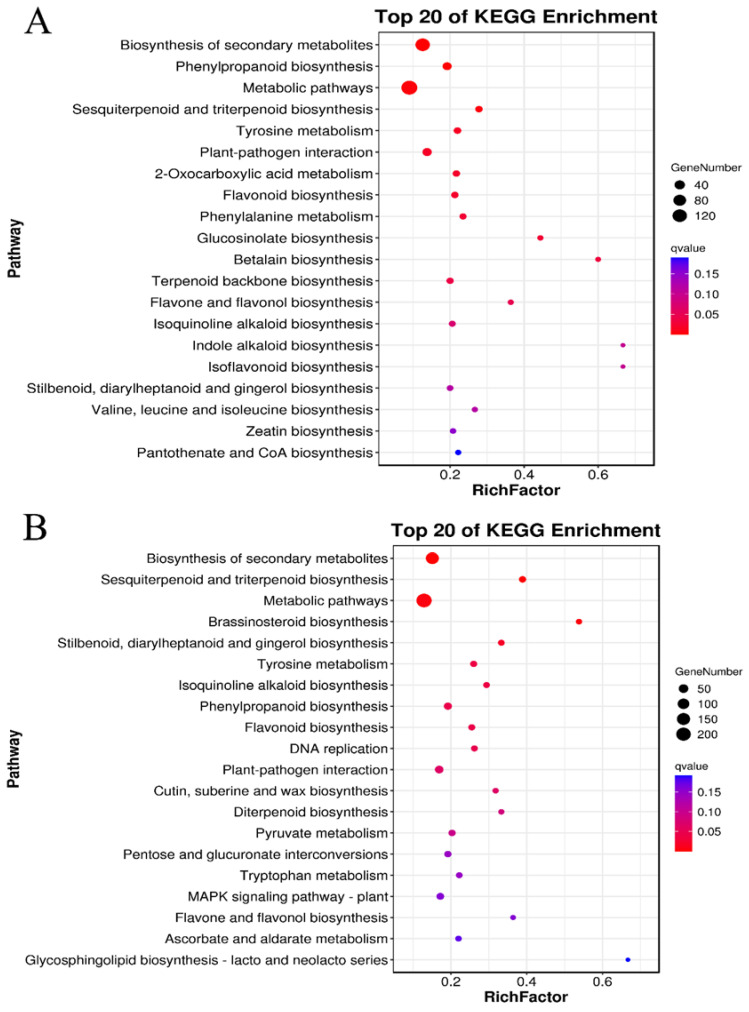
Enrichment Analysis of Differentially Expressed Genes (DEGs) Involved in the Metabolic Pathways Altered in Treated Leaves and Stems via the KEGG Database. (**A**,**B**), two charts show enriched metabolisms altered in treated leaves (C-Y) (**A**) and stems (C-J) (**B**). The KEGG enrichment bubble diagrams indicate that the top 20 enriched metabolisms with the smallest Q value. The ordinate includes plant metabolisms. The abscissa is the enrichment factor (the number of DEGs in the pathway divided by all the numbers in the pathway). Bubble size indicates the quantity. The color code from blue to red indicates the trend from the biggest to smallest Q values. KEGG, Kyoto Encyclopedia of Genes and Genomes.

**Figure 6 ijms-23-12025-f006:**
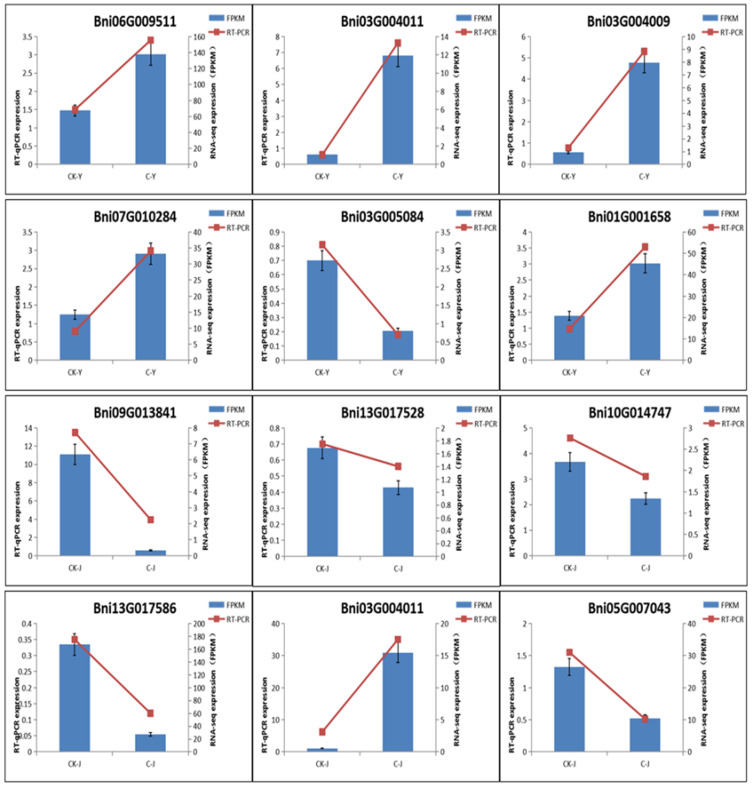
Validation of the Differentially Expressed Genes (DEGs) in GA_3_ and ETH-treated Leaves and Stems. qRT-PCR was Performed for Six DEGs from Leaves and the Other Six from Stems. The results from both FPKM and qRT-PCR were consistent.

**Figure 7 ijms-23-12025-f007:**
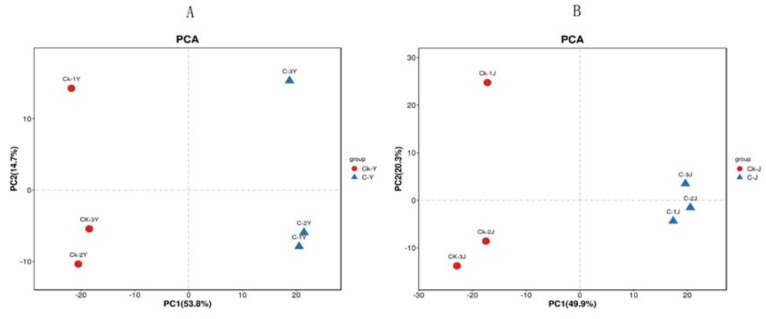
Plots of Principle Component Analysis Showing Metabolic Alterations by the GA_3_ and ETH Treatment. Two plots show differentiations of 99 metabolites in treated (C-Y) vs. control (CK-Y) leaves (**A**) and of 88 metabolites in treated (C-J) vs. control (CK-J) stems (**B**).

**Figure 8 ijms-23-12025-f008:**
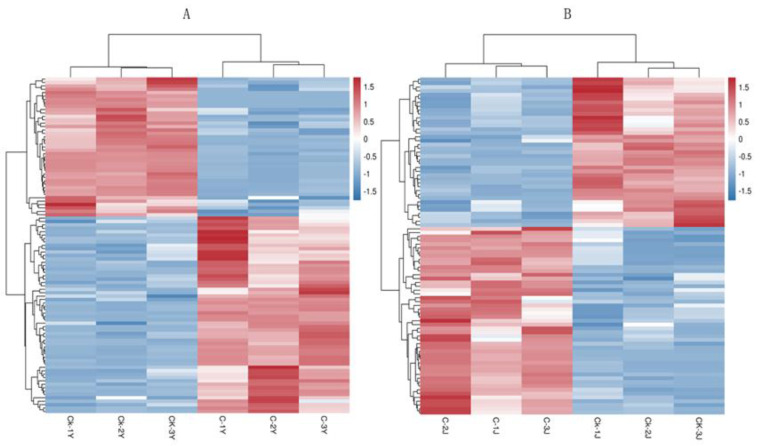
Heatmaps Showing Metabolite Differentiation between Treated and Control Samples. (**A**) Differential profiles of 99 metabolites in the treated (C-Y) and control (CK-Y) leaves; (**B**) differential profiles of 88 metabolites in the treated (C-J) and control (CK-J) stems. Red indicates a high abundance, while blue indicates a low abundance.

**Figure 9 ijms-23-12025-f009:**
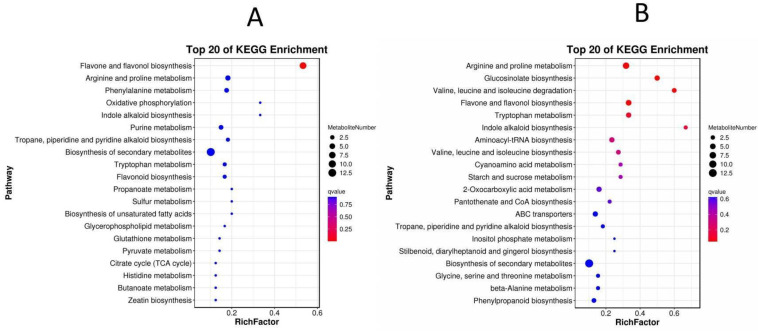
Pathway Mapping and Enrichment that Resulted from Differentially Accumulated Metabolites (DAMs) Analyzed in KEGG. (**A**) The top 20 metabolisms that resulted from 99 DAMs were enriched via KEGG analysis to show the differentiation between the treated (C-Y) and control (CK-Y) leaves. (**B**) The top 20 metabolisms that resulted from 88 DAMs were enriched via a KEGG analysis to show the differentiation between the treated (C-J) and control (CK-J) stems. The ordinate shows the metabolisms. The abscissa shows the enrichment factor, which is the number of differential metabolites divided by all the numbers in the pathway. The bubble sizes indicate the number of metabolites. The trend of colors from blue to read indicates the Q values. KEGG, Kyoto Encyclopedia of Genes and Genomes.

**Figure 10 ijms-23-12025-f010:**
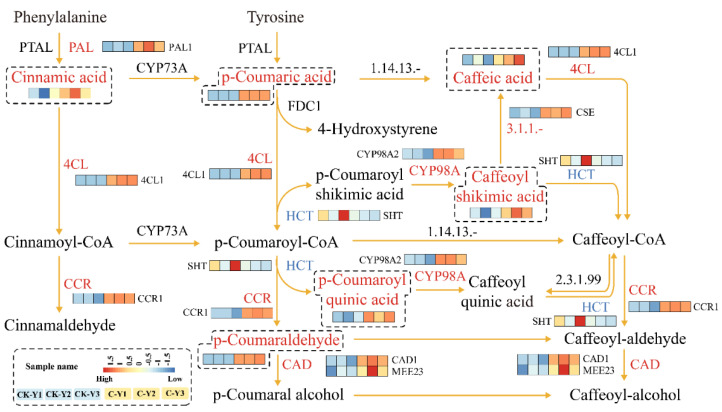
A Schematic Diagram Developed from the Integration of Differentially Expressed Genes (DEGs) and Differentially Accumulated Metabolites (DAMs) in the Phenylpropanoid Pathway in Leaves. Three biological control leaf samples (CK-Y1, CK-Y2, and CK-Y3) and three biological treated leaf samples (C-Y1, C-Y2, and C-Y3) are coded by three squares or rectangles in the order of CK-Y1, CK-Y2, CK-Y3, C-Y1, C-Y2, and C-Y3. The six squares or rectangles were used to create heatmaps. The expression levels of genes and accumulation levels of metabolites in C-Y1, C-Y2, C-Y3, C-Y1, C-Y2, and C-Y3 are coded by colored heatmaps of the six squares or rectangles. PAL, 4CL, CCR, CYP98A, and CAD coded with a red color on each arrow mean that their gene expression levels are upregulated in the treated leaves. HCT coded by a light blue means that its gene expression is reduced. Six metabolite names, which are coded with a red color at end of arrows and included together in a heatmap contained in a dashed box, mean that their levels are increased in the treated leaves.

**Figure 11 ijms-23-12025-f011:**
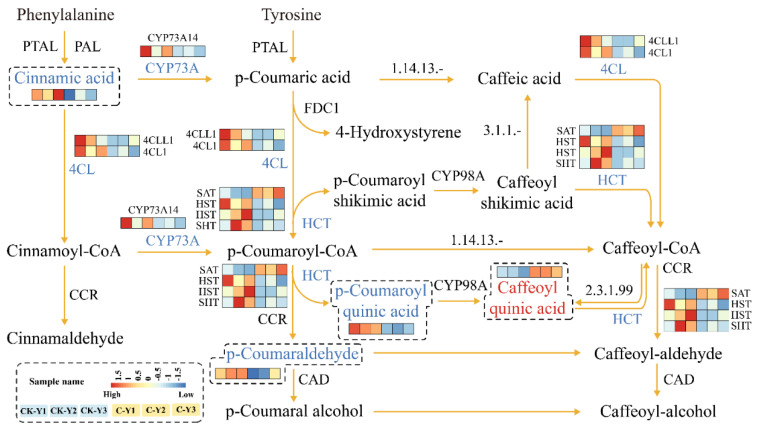
A Schematic Diagram Developed from the Integration of Differentially Expressed Genes (DEGs) and Differentially Accumulated Metabolites (DAMs) in the Phenylpropanoid Pathway in Stems. Three biological control leaf samples (CK-Y1, CK-Y2, and CK-Y3) and three biological treated leaf samples (C-Y1, C-Y2, and C-Y3) are coded by three squares or rectangles in the order of CK-Y1, CK-Y2, CK-Y3, C-Y1, C-Y2, and C-Y3. The six squares or rectangles were used to create heatmaps. The expression levels of genes and accumulation levels of metabolites in C-Y1, C-Y2, C-Y3, C-Y1, C-Y2, and C-Y3 are coded by colored heatmaps of the six squares or rectangles. 4CL, CYP73A, and HCT coded with a light blue color on each arrow mean that their gene expression levels are downregulated in the treated stems. Cinnamic acid, p-coumaraldehyde, and p-coumaroyl quinic acid, which are coded with a light blue color and included together in a heatmap contained in a dashed box, mean that their levels are decreased in the treated stems, while caffeoyl quinic acid coded by a red color means its decrease.

**Table 1 ijms-23-12025-t001:** Throughput and Quality Summary of the RNA Sequences.

Sample	Raw Data (bp)	Clean Data (bp)	Q20 (%)	Q30 (%)	GC (%)
CK-1Y	6584247300	6541109091	97.46%	93.08%	48.18%
CK-2Y	7055284500	7006792142	97.38%	92.95%	48.27%
CK-3Y	5673744900	5635597101	97.36%	92.87%	48.35%
C-1Y	7511753100	7457357511	97.43%	93.09%	48.12%
C-2Y	8558922600	8497437801	97.48%	93.20%	48.19%
C-3Y	8165506500	8109668845	97.53%	93.30%	47.75%
CK-1J	7492708500	7440545683	97.55%	93.36%	48.08%
CK-2J	6456537600	6410364400	97.44%	93.10%	47.86%
CK-3J	5782951500	5745941703	97.46%	93.15%	47.89%
C-1J	7434733800	7384220008	97.35%	92.90%	48.03%
C-2J	7795367100	7742492478	97.37%	92.92%	48.00%
C-3J	6811894200	6768241716	97.49%	93.17%	47.99%

## Data Availability

Not applicable.

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
