# Peer review of "Transcriptional and Metabolic Characterization of Feeding Ramie Growth Enhanced by a Combined Application of Gibberellin and Ethrel"

_ijms, 2022, doi:10.3390/ijms231912025_

Round 1

Reviewer 1 Report

1.     In line number, 21-23 what part of the plant are you comparing with? Mention about that part and rephrase the sentence.

2.     Rephrase the line no. 117-118.

3.     In line, 130-131 what is the level of significance that you have set as threshold?

4.     Submit the raw reads to SRA-NCBI and mention the accession number.

5.     Mention the number of technical and biological replicates used for sequencing. Data from the experiments with single tissue and no replicates are not reliable.

6.     Mention the tools used to check the QC, assembly and trimming of reads in results.

7.     Remove excess spacing after figure 3B.

8.     In line 581, how is it possible to set a reaction volume of 20mL in a PCR machine? If it is μl please correct it.

9.     Were the primers validated with the reverse transcribed RNA products (cDNA)?

10.   Why lignin and flavonoids were selected as an interest? Why other metabolites weren’t chosen?

11.   Images are not clear re-draw the tables represented in figure 4 both A and B.

Author Response

Comments and Suggestions for Authors

  1. In line number, 21-23 what part of the plant are you comparing with? Mention about that part and rephrase the sentence.

Response:

Thank you for your suggestion.

Revised as “1,562 and 2,364 differentially expressed genes (DEGs) were obtained from leaves () and stems (treated versus control), respectively.  Meanwhile, 99 and 88 differentially accumulated metabolites (DAMs) were annotated from treated versus control leaves and treated versus stems  in, respectively. ”

  1. Rephrase the line no. 117-118.

Response:

Revised as “The effects of the GA3 and ETH treatment on the contents of lignin and total flavonoids in the leaves and stems of ramie were studied.”

  1. In line, 130-131 what is the level of significance that you have set as threshold?

Response:

Revised as “In addition,, these data showed that the contents of total flavonoids in the stems slightly responded to the GA3 and ETH treatments.”

Significance was accepted at P-values < 0.05.

  1. Submit the raw reads to SRA-NCBI and mention the accession number.

Response:

Thank you for your suggestion. We are totally agreement with you. In the present study, all of the data has been presented on the attachments. We are analyse other pathway according to the result of this Omics, which cannot submitted to NCBI as soon as possible. If there are any question, please request the data by the corresponding author through email ( ibfcjyc@vip.sina.com).

  1. Mention the number of technical and biological replicates used for sequencing. Data from the experiments with single tissue and no replicates are not reliable.

Response:

There are 3 biological replicates were used for sequencing. No technical replicates was consider in the present study. Thank you for your comment. We will double consider your comments on the technical replicates in our further study.

  1. Mention the tools used to check the QC, assembly and trimming of reads in results.

Response:

Thank you for your suggestion.

The QC was described on the material and methods, section 4.3.

For RNA sequence data analysis, the original sequencing data were first analyzed by FastQC, and low-quality numerical readings were filtered and discarded by Trimmomatic version 0.36.

  1. Remove excess spacing after figure 3B.

Response:

Revised.

  1. In line 581, how is it possible to set a reaction volume of 20mL in a PCR machine? If it is μl please correct it.

Response:

Thank you for your comments.

Corrected.

  1. Were the primers validated with the reverse transcribed RNA products (cDNA)?

Response:

Thank you for your comment.

The primers were validated with the reverse transcribed RNA products (cDNA)

  1. Why lignin and flavonoids were selected as an interest? Why other metabolites weren’t chosen?

Response:

Because lignin secondary metabolites and flavonoid share the same precursors, the synthesis of secondary metabolites that share the phenylpropanoid metabolic pathway with hormones will also be affected during the process of regulating the lignin metabolic pathway.

  1. Images are not clear re-draw the tables represented in figure 4 both A and B.

Response:

Revised.

Reviewer 2 Report

The research article entitled "Research on the Transcriptomics and Metabolomics to Analyze the Molecular Mechanism of Feeding Ramie Growth Regulation with Combined Application of Gibberellin and Ethrel" by Hongdong Jie et al., for the first time provides an in-depth analysis of the difference of lignin and other secondary metabolites content between control and Gibberellin (GA3) and ethylene (ETH) theated ramie (Boehmaria nivea L.).

The introduction provides all the necessary background information and clearly defines the aim of the study. Results are clearly presented, sound and convincing. Materials and methods are adequately presented with the exception of the bio-informatics part:

1. It is stated (Line 530) that the reads were mapped on a genome of tea plants (Camellia sinensis) – why this certain genome was used? Would it be better to either use a de-novo transcriptome assembly approaches or use the genome sequence of the ramie is available at NCBI (https://www.ncbi.nlm.nih.gov/data-hub/genome/GCA_018132145.1/). Please explain.

2. The level of gene expression was estimated using the FPKM method, why TPM was not used here? With the case of TPMs the sum of TPMs in each sample are the same and it is easier to compare the proportion of reads that mapped to a gene in each sample while with FPKM the sum of the normalized reads in each sample may be different, making it harder to directly compare samples.

3. At line 558 the Perl script for removing the adducts, dimers, de-isotoping and removing the in-source fragments was used, I think it would be beneficial to upload that script to github or other on-line repository and provide an url to it.

4. Please specify the acquisition method for tandem mass spectrometry (MS/MS) and it’s parameters – was it data dependent acquisition (DDA) or data independent acquisition (DIA).

There are also some minor formatting issues that need to be corrected:

1. The data availability statement is not filled in. I think it would be beneficial to upload the raw sequence data to the SRA archive and provide a link here.

2. The references section starts with the reference “2” (line 611), which is in fact the fererence “1”. At the end of references there is an additional blank reference 49 (line 712)

I think this article is of interest to the readers and should published after a minor revision addressing my concerns and aforementioned formatting issues.

Author Response

The research article entitled "Research on the Transcriptomics and Metabolomics to Analyze the Molecular Mechanism of Feeding Ramie Growth Regulation with Combined Application of Gibberellin and Ethrel" by Hongdong Jie et al., for the first time provides an in-depth analysis of the difference of lignin and other secondary metabolites content between control and Gibberellin (GA3) and ethylene (ETH) theated ramie (Boehmaria nivea L.).

The introduction provides all the necessary background information and clearly defines the aim of the study. Results are clearly presented, sound and convincing. Materials and methods are adequately presented with the exception of the bio-informatics part:

I think this article is of interest to the readers and should published after a minor revision addressing my concerns and aforementioned formatting issues.

Response:

We appreciate your time and comments on our manuscript.

  1. It is stated (Line 530) that the reads were mapped on a genome of tea plants (Camellia sinensis) – why this certain genome was used? Would it be better to either use a de-novo transcriptome assembly approaches or use the genome sequence of the ramie is available at NCBI (https://www.ncbi.nlm.nih.gov/data-hub/genome/GCA_018132145.1/). Please explain.

Response:

Thank you.

We double checked the data and revised as “Reads contaminated by adapter sequences were cut, and then STATR version 2.5.3a software was used to map the clean reads to reference genome of ramie (Boehmaria nivea L.)” .

  1. The level of gene expression was estimated using the FPKM method, why TPM was not used here? With the case of TPMs the sum of TPMs in each sample are the same and it is easier to compare the proportion of reads that mapped to a gene in each sample while with FPKM the sum of the normalized reads in each sample may be different, making it harder to directly compare samples.

Response:

Thank you for your suggestion.

FPKM method and TPM method are the classical method in gene expression. In addition, some references used FPKM method in gene expression [1, 2].

Reference:

[1] Wang, C., Dong, Y., Zhu, L. et al. Comparative transcriptome analysis of two contrasting wolfberry genotypes during fruit development and ripening and characterization of the LrMYB1 transcription factor that regulates flavonoid biosynthesis. BMC Genomics 21, 295 (2020). https://doi.org/10.1186/s12864-020-6663-4

[2] [1] Zhang Q ,  Wang L ,  Liu Z , et al. Transcriptome and metabolome profiling unveil the mechanisms of Ziziphus jujuba Mill. peel coloration[J]. Food Chemistry, 2019, 312:125903. 10.1016/j.foodchem.2019.125903

  1. At line 558 the Perl script for removing the adducts, dimers, de-isotoping and removing the in-source fragments was used, I think it would be beneficial to upload that script to github or other on-line repository and provide an url to it.

Response:

Thank you for your suggestion.

In the present study,these data was done on the platform of an ultra-fast all-in-one FASTQ preprocessor according to the described by Chen et al.(2018).

  1. Please specify the acquisition method for tandem mass spectrometry (MS/MS) and it’s parameters – was it data dependent acquisition (DDA) or data independent acquisition (DIA).

Response:

Thank you for your suggestion.

LIT and triple quadrupole (QQQ) scans were acquired on a triple quadrupole-linear ion trap mass spectrometer (Q TRAP), API 6500 Q TRAP LC/MS/MS System, equipped with an ESI Turbo Ion-Spray interface, operating in a positive ion mode and controlled by Analyst 1.6.3 software (AB Sciex). The ESI source operation parameters were as follows: ion source, turbo spray; source temperature 500°C; ion spray voltage (IS) 5500 V; ion source gas I (GSI), gas II(GSII), curtain gas (CUR) were set at 55, 60, and 25.0 psi, respectively; the collision gas(CAD) was high. Instrument tuning and mass calibration were performed with 10 and 100 μmol/L polypropylene glycol solutions in QQQ and LIT modes, respectively. QQQ scans were acquired as MRM experiments with collision gas (nitrogen) set to 5 psi. DP and CE for individual MRM transitions was done with further DP and CE optimization. A specific set of MRM transitions were monitored for each period according to the metabolites eluted within this period. DIA or DDA method is not used.

There are also some minor formatting issues that need to be corrected:

  1. The data availability statement is not filled in. I think it would be beneficial to upload the raw sequence data to the SRA archive and provide a link here.

Response:

Thank you for your suggestion. We are totally agreement with you. In the present study, all of the data has been presented on the attachments. We are analyse other pathway according to the result of this Omics, which cannot submitted to NCBI as soon as possible. If there are any question, please request the data by the corresponding author through email ( ibfcjyc@vip.sina.com).

  1. The references section starts with the reference “2” (line 611), which is in fact the fererence “1”. At the end of references there is an additional blank reference 49 (line 712)

Response:

Thank you for your comments.

Revised.
